# Long-Term Outcomes of Ablative Carbon-Ion Radiotherapy for Central Non-Small Cell Lung Cancer: A Single-Center, Retrospective Study

**DOI:** 10.3390/cancers16050933

**Published:** 2024-02-25

**Authors:** Shuri Aoki, Hitoshi Ishikawa, Mio Nakajima, Naoyoshi Yamamoto, Shinichiro Mori, Masaru Wakatsuki, Noriyuki Okonogi, Kazutoshi Murata, Yuji Tada, Teruaki Mizobuchi, Ichiro Yoshino, Shigeru Yamada

**Affiliations:** 1QST Hospital, National Institutes for Quantum Science and Technology, 4-9-1 Anagawa, Inage-ku, Chiba 263-8555, Japan; aoki.shuri@qst.go.jp (S.A.); nakajima.mio@qst.go.jp (M.N.); n.yamamoto@chouseihp.jp (N.Y.); mori.shinichiro@qst.go.jp (S.M.); wakatsuki.masaru@qst.go.jp (M.W.); okonogi.noriyuki@qst.go.jp (N.O.); murata.kazutoshi@qst.go.jp (K.M.); yamada.shigeru@qst.go.jp (S.Y.); 2Department of Radiology, University of Tokyo Hospital, 3-7-1 Hongo, Tokyo 113-8655, Japan; 3Department of Radiation Oncology, Juntendo University Graduate School of Medicine, 2-1-1 Hongo, Tokyo 113-8421, Japan; 4Department of Pulmonary Medicine, International University of Health and Welfare, Narita Hospital, Hatakeda 852, Chiba 286-8520, Japan; ytada25@iuhw.ac.jp; 5Department of Respiratory Surgery, Social Welfare Organization Saiseikai Imperial Gift Foundation, Chibaken Saiseikai Narashino Hospital, 1-1-1 Izumi-cho, Chiba 275-8580, Japan; tmizobuc@gmail.com; 6Department of Thoracic Surgery, International University of Health and Welfare, Narita Hospital, Hatakeda 852, Chiba 286-8520, Japan; iyoshino@iuhw.ac.jp

**Keywords:** ablative radiotherapy, carbon-ion radiotherapy, central lung tumor, non-small cell lung cancer, proximal bronchial tree, radiation pneumonitis

## Abstract

**Simple Summary:**

Ablative radiotherapy for central early stage non-small cell lung cancer (NSCLC) is controversial in its feasibility and optimal dose prescription due to the risk of severe pneumonia and adverse events associated with mediastinal organs. Carbon-ion radiotherapy (CIRT) is a promising modality of radiotherapy with steep and linear dose distribution and high biological efficacy. We retrospectively analyzed the long-term results of ablative CIRT using 68.4 Gy in 12 fractions for central early stage NSCLC. Irradiation doses to lungs and mediastinal organs such as bronchus and esophagus were controlled and no serious mediastinal organ-related adverse events occurred. In addition, the efficacy was not inferior to that reported in previous studies, including photon SBRT or proton radiotherapy. This means that our CIRT regimen is expected to have a relatively high safety profile for central NSCLC with tumor and organ at risk (OAR) proximity.

**Abstract:**

The aim of this study is to assess the efficacy and safety of ablative carbon ion radiotherapy (CIRT) for early stage central non-small cell lung cancer (NSCLC). We retrospectively reviewed 30 patients who had received CIRT at 68.4 Gy in 12 fractions for central NSCLC in 2006–2019. The median age was 75 years, and the median Karnofsky Performance Scale score was 90%. All patients had concomitant chronic obstructive pulmonary disease, and 20 patients (67%) were considered inoperable. In DVH analysis, the median lung V5 and V20 were 15.5% and 10.4%, and the median Dmax, D0.5cc, D2cc of proximal bronchial tree was 65.6 Gy, 52.8 Gy, and 10.0 Gy, respectively. At a median follow-up of 43 months, the 3-year overall survival, disease-specific survival, and local control rates were 72.4, 75.8, and 88.7%, respectively. Two patients experienced grade 3 pneumonitis, but no grade ≥3 adverse events involving the mediastinal organs occurred. Ablative CIRT is feasible and effective for central NSCLC and could be considered as a treatment option, especially for patients who are intolerant of other curative treatments.

## 1. Introduction

Surgery is the standard treatment for early stage non-small cell lung cancer (NSCLC) [1,2,3]. However, the increasing elderly population with multiple comorbidities also requires minimally invasive alternatives. Stereotactic body radiotherapy (SBRT) has been established as an alternative treatment for patients with inoperable peripheral early stage NSCLC, achieving favorable local control and safety [4,5,6,7]. On the other hand, for central early stage NSCLC, both surgery and SBRT need careful consideration due to the high risk of injury to mediastinal organs and loss of respiratory function [8,9,10,11,12,13,14,15,16,17]. Indeed, SBRT for centrally located NSCLC causes 11 times as many severe AEs as it does for peripheral NSCLC [10]. Therefore, the appropriate dose prescription for central SBRT remains controversial.

Meanwhile, carbon-ion radiotherapy (CIRT) is a type of particle RT using carbon ions that was first clinically applied in the world at QST hospital in Japan. The advantages of particle RT, particularly CIRT, can be summarized mainly in two respects: physical and biological. Physically, CIRT has a peak in the dose spread called the Bragg peak, which can be expanded according to the shape and location of the tumor (spread-out Bragg peak; SOBP). SOBP allows CIRT to deliver sufficient doses focused on the tumor with a few numbers of beams [18,19]. Therefore, a steep dose gradient can be created between the tumor and the mediastinal OAR, which is expected to reduce severe AEs, even in central lung tumors. Biologically, carbon ions also have higher linear energy transfer (LET) values compared to photons and protons, resulting in high relative biological effects (RBE) and cytotoxicity against X-ray resistant tumor cells [20]. In addition, the LET of carbon-ion beams advances to the peak region in the body, resulting in an increased biological effect on deep-located tumors. This is also advantageous from a therapeutic point of view [21,22].

We have been exploring CIRT for lung tumors since 1994 and have developed high-dose CIRT for localized NSCLC, achieving better results relative to photon SBRT [23,24]. Currently, the protocol of 50 Gy (described as a relative biological effect [RBE]-weighted dose based on the modified microdosimetric kinetic model) in a single fraction (fr) is employed for peripheral, early stage NSCLC and provides tumor control comparable to SBRT but with an extremely high safety profile [25]. In contrast, for central NSCLC, specific safety-oriented protocols were developed in 2006, i.e., 68.4 Gy/12 fr, has been used to date but not adequately evaluated. In this study, we evaluated the efficacy and safety of CIRT for the treatment of central NSCLC.

## 2. Materials and Methods

### 2.1. Patients

Patients with central lung tumors who had received CIRT with 68.4 Gy in 12 Fr at QST Hospital in Chiba, Japan, between July 2006 and September 2019 were retrospectively analyzed. All patients were treated according to the institutional protocols for hilar and hilar-adjacent NSCLC with extrabronchial masses. This study was approved by the hospital Ethics Committee (approval number: N22-023), and written informed consent was obtained from all patients.

Inclusion criteria were as follows: (1) Japanese patients with pathologically confirmed NSCLC, (2) stage T1-T2N0M0 (International Union for Cancer Control; UICC, 7th edition) tumors ≤7 cm without lymph node or distant metastasis, and (3) World Health Organization performance status of 0 to 2. Central lesions were defined as tumors within 2 cm of the proximal bronchial tree (PBT) or immediately adjacent to the mediastinal or pericardial pleura, as described in the Radiation Therapy Oncology Group Trial (RTOG) 0813 [15]. Ultra-central lung tumors (i.e., tumors invading the hilar structures and bronchial tree proximal to the main bronchus) were excluded from the present study as they were not considered safe targets for this regimen [26,27,28,29], and were treated with a reduced dose of 48 Gy in 12 Fr. Patients receiving concurrent chemotherapy and those with prior irradiation in the same area were also excluded. The pretreatment NSCLC stage was determined based on the UICC for International Cancer Control 7th edition using thin-slice computed tomography (CT) and positron emission tomography (PET)-CT. In the retrospective analysis of the clinical data, the stage was re-evaluated based on the 8th edition. No patients were excluded due to respiratory function.

### 2.2. Treatment Planning

CIRT treatment planning was performed using four-dimensional (4D) CT at 2.5 mm or 5.0 mm intervals. Patients were immobilized in the supine or abdominal position on a body frame under free-breathing conditions using an external respiratory monitoring system (PSM1520; Toyonaka Lab, Osaka, Japan for 2015–2016; AZ-733 V^®^; Anzai Medical, Tokyo, Japan for 2016). The HIPLAN (National Institute for Radiological Science, Chiba, Japan) and Xio-N (ELECTA; Stockholm, Sweden; Mitsubishi Electric, Tokyo, Japan) systems were used for CT planning until July 2013. A respiratory gating system was developed to minimize the effects of breath-induced tumor motion.

The gross tumor volume (GTV) was delineated, including the lung tumor at the lung window (1600 Hounsfield units (HU); level, −300 HU). The clinical target volume (CTV) was defined as the GTV plus a 0.5 to 1.0 cm margin. In addition, we defined the beam field-specific target volume (FTV) by extending the 3D treatment planning technique to 4D [30] and a 2 to 3 mm setup margin to create the planning target volume (PTV). Major OARs (heart, lungs, esophagus, spinal cord, and proximal tracheobronchial tree) were contoured according to the RTOG 0236 guidelines [28,31].

The available treatment angles were increased by tilting the treatment bed up to ±20° for beams delivered horizontally or vertically, and several different angular ports were selected and combined for OAR protection. CIRT planning and irradiation details at our institution have been previously reported [23,24,32,33]. The carbon-ion dose is expressed in Gy and calculated by multiplying the physical dose by the RBE [34]. A fixed dose of 68.4 Gy was delivered in 12 Fr (i.e., four consecutive days per week over three consecutive weeks). The total dose was prescribed to the isocenters and adjusted to cover the PTV with a 95% isodose line of the prescribed dose. Dose constraints for organs at risk (OARs) were strictly adhered to, with spinal (maximum point dose; Dmax) < 28 Gy, esophageal (dose to the hottest 0.2cc; D0.2cc) < 50 Gy, and mainstem bronchus (D2cc) < 50 Gy, taking priority over target coverage. No dose constraints on the lung doses were defined. Figure 1 shows examples of the CIRT treatment plans.

### 2.3. Follow-Up and Statistical Analysis

Follow-up after CIRT consisted of a physical examination and a contrast-enhanced CT performed every 3 months for the first 2 years and at least every 6 months thereafter. Local recurrence was defined as progressive and increased CT abnormality. If local recurrence was suspected, ^18^F-fluorodeoxyglucose (FDG) tomography/computed tomography and a biopsy were performed. Acute and late AEs were assessed according to the Common Terminology Criteria for Adverse Events Version 5.0 [35]. AEs occurring within 3 months were defined as acute AEs, and those occurring thereafter were defined as late effects. The dosimetric evaluation was performed using dose-volume histogram (DVH) analysis to verify treatment quality. Doses to the OARs were calculated for the following structures: for PBT/esophagus/spinal cord, maximum dose to a point (Dmax) or small volumes (D0.2/0.5/1/2cc), for lungs, mean lung total dose (MLD total) and percentage of lung volume irradiated above 5/20 Gy (lung V5/V20).

For categorical variables, between-group comparisons were performed using Pearson’s χ^2^ test. Overall survival (OS), disease-specific survival (DSS), and local control (LC) rates were calculated from the first day of CIRT until the event or last patient contact using Kaplan–Meier curves. Statistical analyses were performed using R software (https://www.r-project.org/, accessed on 15 January 2024), and statistical significance was set at *p* < 0.05.

## 3. Results

### 3.1. Patients

A total of 30 patients with a median age of 75 (range 55–85) years who had been treated with 68.4 Gy in 12 Fr were evaluated (Table 1). The median Karnofsky Performance Scale (KPS) score was 90% (range 70–100%). All patients had concomitant chronic obstructive pulmonary disease (COPD). According to the Chronic Obstructive Lung Disease (COLD) spirometry grade [36], the numbers of patients with grades 1, 2, 3, and 4 were 11, 14, 5, and 1, respectively, with six cases (20%) ≥ grade 3. Twenty (67%) patients were deemed inoperable. The median follow-up time was 42 (range 8–163) months for all patients and 63 months for the surviving patients.

### 3.2. Treatment Planning

DVH was available for analysis in 26 patients, as the dose data were partially missing for the remaining four patients. Median GTV volume was 22.4 (0.9–98.6) cm^3^. The GTV dose was within the prescribed range of ±5% in all cases. For the lungs, median lung V5 and V20 were 15.5 (10.2–24.1) % and 10.4 (3.7–18.2) %, respectively. The MLD total was 5.6 (2.6–10.9) Gy. PBT-tumor distance was 2 (0–18) mm, median Dmax of PBT was 65.6 (3.8–71.2) Gy, median D0.2cc, D0.5cc, D1cc, and D2cc were 58.8 (2.8–70.7) Gy, 52.8 (2.7–69.2) Gy, 39.5 (2.4–68.6) Gy, and 10.0 (0.2–51.6) Gy, respectively. The median of the maximum dose to the PBT in the 8 patients with tumors touching the PBT was 67.8 (61.8–70.6) Gy, and in the 16 patients with tumors at 1 to 15 mm from the PBT, it was 61.4 (50.5–71.2) Gy. In contrast, in the remaining 2 patients with tumor distances of 18–20 mm from the PBT, the maximum PBT doses were extremely low at 3.8 and 1.9 Gy, respectively. The median maximum dose to the esophagus was 9.9 (1.8–60.7) Gy. In the highest case, where the distance to the GTV was 8.6 mm and that to the CTV was 2.0 mm, the maximum dose point was 60.7 Gy, which was 5.4 Gy in the D1cc setting (Figure 2). In contrast, the maximum dose to the esophagus was <20 Gy in 21 of the 23 cases with a distance to the tumor of ≥20 mm, and <10 Gy in all 12 cases with a distance to the tumor of ≥30 mm. The median D0.2cc, D0.5cc, and D1cc into the esophagus were 4.2 (1.4–37.5), 3.6 (0.8–30.9), and 3.0 (0.7–23.5) Gy, respectively. The doses delivered to the main OARs are listed in Table 2.

### 3.3. Treatment Effects

At the last follow-up, 14 patients (47%) had died, of which two were cause-specific deaths. Among the other 12, nine deaths were lung-related but treatment-unrelated (e.g., pneumonitis), 2 were cancer-related but without the recurrence of lung cancer, and 1 was from heart failure. The median OS was 77 months (range 41–NA). The 1-, 3-, and 5-year OS rates were 96.7% (95% CI: 78.6–99.5%), 72.4% (95% CI: 52.3–85.2%), and 63.3% (95% CI: 42.0–78.6%), respectively (Figure 3A). Seven patients (23%) had recurrences; three had local recurrence, five had non-local recurrences, and one had both. The five non-local recurrences included two patients with intrathoracic metastases (intrapulmonary or mediastinal lymph nodes), one with clavicular fossa lymph node metastasis, and two with distant organ metastasis (brain, bone, and liver). The 1-, 3-, and 5-year DSS rates were 83.2% (95% CI: 64.2–92.6%), 75.8% (95% CI: 64.2–92.6%), and 75.8% (95% CI: 55.7–87.7%), respectively (Figure 3B). The 1-, 3-, and 5-year LC rates were 93.1% (95% CI: 75.1–98.2%), 88.7% (95% CI: 68.5–96.2%), and 88.7% (95% CI: 68.5–96.2%), respectively (Figure 3C).

The results of the univariate analyses of the factors predicting OS are summarized in Table 3. Age (≤75 vs. >75 years, *p* = 0.047) and histology (squamous vs. other, *p* = 0.015) were significant prognosticators. In contrast, the KPS score, cancer stage, and COPD severity, which are known to affect life expectancy, were not significant factors.

### 3.4. Adverse Events

In the acute phase, grade 2 pneumonitis was observed in three patients (10%), and in the late phase, grade 2 and 3 pneumonitis developed in two cases (6.7%), respectively. The two grade 3 cases had stage 3 COPD prior to treatment and were assessed and treated for their symptoms rather than radiological findings. Grade 2 bronchial stenosis with wheezing and coughing occurred in one other case. There were no cases of grade ≥2 esophagitis, but grade 1 occurred in two patients (6.7%); of these, one had received the highest esophageal dose, 60.7 Gy. No grade ≥4 AEs occurred, and we did not observe serious AEs specific to central irradiation (e.g., hemoptysis, tracheoesophageal perforations) during the long-term observation period. Acute and late AEs are summarized in Table 4.

## 4. Discussion

### 4.1. Radiation Therapy for Central NSCLC

The usefulness of SBRT for peripheral early stage NSCLC has been confirmed [4,5,6,7,37,38]; however, its use in treating central lesions remains controversial due to the increased risk of serious AEs [10,11]. Central tumors are independent risk factors for severe AEs even when ultra-central tumors are excluded, and researchers have documented the high toxicity of central SBRT [10,12,14,15,39,40,41,42,43,44,45]. Table 5 summarizes reports on hypo fractionated RT for central lung tumors, mainly SBRT, for the treatment of NSCLC. Severe pneumonitis is the most common AE after central chest irradiation [10,14,40,42,43], grade 5 hemoptysis, tracheoesophageal perforation, and cardiovascular events have often been reported [10,14,39,40,41,42,43,44,45,46]. These AEs are specific to central SBRT with a high dose to the mediastinal organs; although rare, their onset can be sudden and difficult to predict and treat.

### 4.2. Tumor Control

Although there is no established dose prescription for photon SBRT to control stage I NSCLC, BED10 ≥ 100 Gy is considered a significant predictor of LC [42,47,48]. Results from a multicenter study in Japan [48] showed that SBRT for stage I NSCLC with BED10 ≥ 100 Gy significantly improved 3-year OS and LC. Another study reported a 2-year LC rate of 94% for BED10 ≥ 100 Gy and 80% for BED10 < 100 Gy (*p* = 0.02) [42]. While lower dose prescriptions are often used in central NSCLC due to the risk of AEs, SBRT using BED10 ≥ 100 Gy has been reported to provide acceptable tumor control [10,14,15,41].

To the best of our knowledge, there are no reports of ablative CIRT for central early stage lung cancer. However, there have been several reports on similar irradiation using proton beams. Kanemoto et al. [49] reported the results of PBT for stage I NSCLC using 72.6 Gy in 22 Fr (BED10 96.6 Gy) for central tumors and 66 Gy in 10–12 Fr (BED10 102.3–109.56 Gy) for peripheral tumors. The 3-year LC rates were 63.9% and 88.4%, respectively, suggesting insufficient dosing for central NSCLCs. Nakamura et al. [50] retrospectively studied cases of hypo fractionated proton beam therapy for central early stage lung cancer and reported that patients who received less than 110 Gy of BED10 had a lower progression-free survival. As with SBRT, it may be that a dose of BED10 ≥ 100 Gy is required for tumor control in proton beam therapy. Assuming the same estimation as for SBRT, the dose prescription in the present study was a uniform 107.4 Gy in BED10, a dose that is expected to provide adequate tumor control. Indeed, tumor control in this study was comparable to that reported for SBRT using a BED10 ≥ 100 Gy with regard to long-term results. Considering that the present study included relatively large tumors of T2 and T3 (UICC 8th), accounting for 60% and 13% of the cases, respectively, these results seem encouraging. Age (≤75 years) and pathology (squamous cell carcinoma) were identified as prognostic factors for OS in the univariate statistics, consistent with previous reports [51,52]. The absence of effects on tumor size and COPD severity may be attributable to the small sample size.

The risk of AEs will continue to make it difficult to prescribe a significant dose escalation for central lung tumors. However, as shown in a recent paper by Chang JY et al., SBRT combined with ICI has been reported to prolong event-free survival, even in Stage I NSCLC [53]. To improve prognosis, combined ICI could be a promising approach in the future, rather than increasing dose prescription or treatment margins of radiotherapy itself.

### 4.3. CIRT Safety

We identified two major clinical safety advantages of ablative CIRT for central early stage NSCLC. The first is the lower risk of radiation pneumonitis; the sharp dose distribution of CIRT has the advantage of decreased dosing to normal lung tissue [23,54]. Indeed, the doses to the lungs in the present study were sufficiently low across all indices, with notable differences from previous SBRT studies, particularly the mean lung dose (MLD) and lung V5 [14,55,56,57]. In recent years, in addition to lung V20 [51,52], low-dose lung areas, evaluated using lung V5, 10, etc., and MLD have received increased attention as indicators of dose-associated clinical pneumonitis [14,55,56,57]. The reduced low-dose lung area may have led to lower toxicity. However, previous reports of RT for central lung tumors suggest that even if the irradiation area is similarly limited, central radiation pneumonitis with stenosis in BPT tends to be severe. In the present study, two cases of grade 3 pneumonitis were observed, indicating a higher risk compared to that associated with CIRT for peripheral early stage lung cancers. However, most patients had interstitial pneumonitis or lung dysfunction that made other treatments difficult, which may also be related to the occurrence of severe pneumonitis. The second and greatest benefit of CIRT is the reduced risk of damage to mediastinal regions, such as the trachea, esophagus, and cardiovascular organs. In these organs, even small areas of high-dose irradiation can cause injuries, such as perforation, ulceration, and bleeding, which are rare but fatal AEs with sudden onset and poor predictability. In this study, no serious AEs occurred in the mediastinal organs during the long-term follow-up period. Although the sample size was small, these results indicate the safety of central CIRT.

Compared to photon SBRT, which uses multiportal and rotational techniques, CIRT allows for sufficient dose control with fewer beams, creating a linear dose gradient between the tumor and mediastinal OARs. As a result, it is possible to keep these OAR doses low even in central tumors with less than 2 cm between the tumor and the bronchial tree or other mediastinal organs. In previous reports, several dose parameters such as Dmax, D0.5cc, and D2cc have been used to describe the association between mediastinal OAR dose, AEs, and survival, with some reports using relatively large volume indicators such as D10cc [45,58,59]. Their views on volume effects on mediastinal organs suggest that a reduction in high dose irradiated volumes could lead to fewer serious AEs. The HILUS study concluded that D0,2cc, D0.5cc, and D1cc of the main bronchus plus trachea was most associated with grade 5 toxicity [29]. Farrugia M. et al. reported that D4cc (>18 Gy in 5 fr) for PBT correlated most with worse non-cancer-related survival [59]. In the present study, we evaluated OAR doses using Dmax, D0.2cc, D0.5cc, D1cc, and D2cc, and found that D > 0.5 cc, which is considered clinically important, was well controlled even in cases with high Dmax. This means that in CIRT, even in cases where the OAR is in proximity/contact with the target and partly exposed to high doses, it could be possible to reduce the extension of the high dose area, thereby avoiding serious AEs. Although there are no data directly comparing CIRT with other treatment modalities, a dosimetric report on the superiority of CIRT supports the present results [60,61]. Of course, the larger number of fractions than most reports on SBRT may have contributed to the safety profile of the present study, but at least in the treatment of central NSCLC, the CIRT regimen performed at our institution would be expected to have a relatively high safety profile. However, the advantages of CIRT may not be fully exploited for ultra-central tumors, particularly those touching or invading the esophagus or trachea, and the optimal dose prescription requires more careful exploration [29,30,46].

### 4.4. Limitations

The present study had several limitations. First, this was a retrospective study conducted at a single institution. Second, the small sample size may have limited the statistical power. Third, our ongoing treatment regimen development resulted in minor modifications to the irradiation protocol within the case series. Furthermore, methods for comparing the biological effects of CIRT and photon SBRT have not yet been established, and it remains difficult to compare data across institutions. Future basic and multicenter studies are required to explore the role of appropriate doses of CIRT in treating central NSCLC.

## 5. Conclusions

We reported the results of ablative CIRT (68.4 Gy in 12 Fr) for central NSCLC. CIRT is considered to be a safe and effective treatment for central NSCLC, as dose reduction to intrathoracic OARs reduces the risk of fatal AEs. Further research to clarify the most effective indications and dose constraints for CIRT could make a significant contribution to the treatment of central NSCLC.

## Figures and Tables

**Figure 1 cancers-16-00933-f001:**
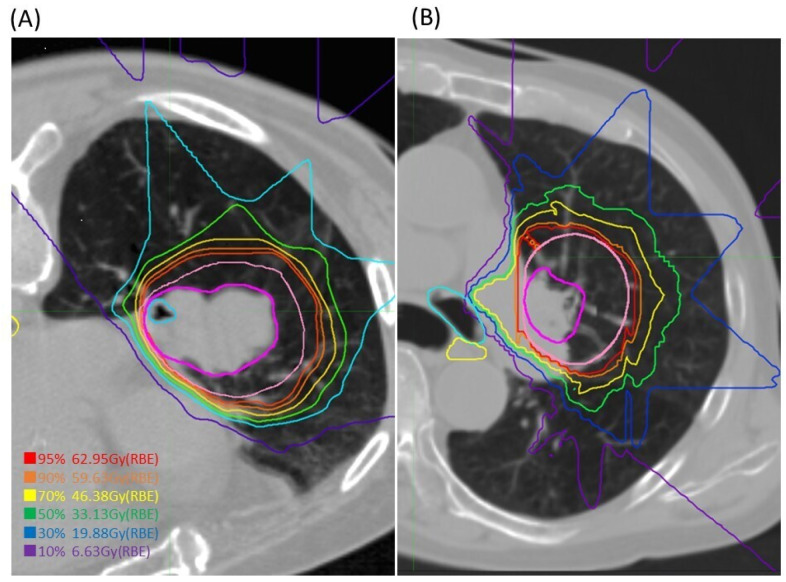
Dose distributions of CIRT for central NSCLC. The red, orange, yellow, green, blue, and purple lines show doses of 95%, 90%, 70%, 50%, 30%, and 10%, respectively; GTV was drawn in magenta, and CTV in pale pink. In cases where the OAR (in this case the lobe bronchus) is within the tumour, the OAR dose is sometimes as high as the tumour (**A**), but if there is a small distance from the tumour, the OAR can be successfully avoided by adjusting the beam direction (**B**).

**Figure 2 cancers-16-00933-f002:**
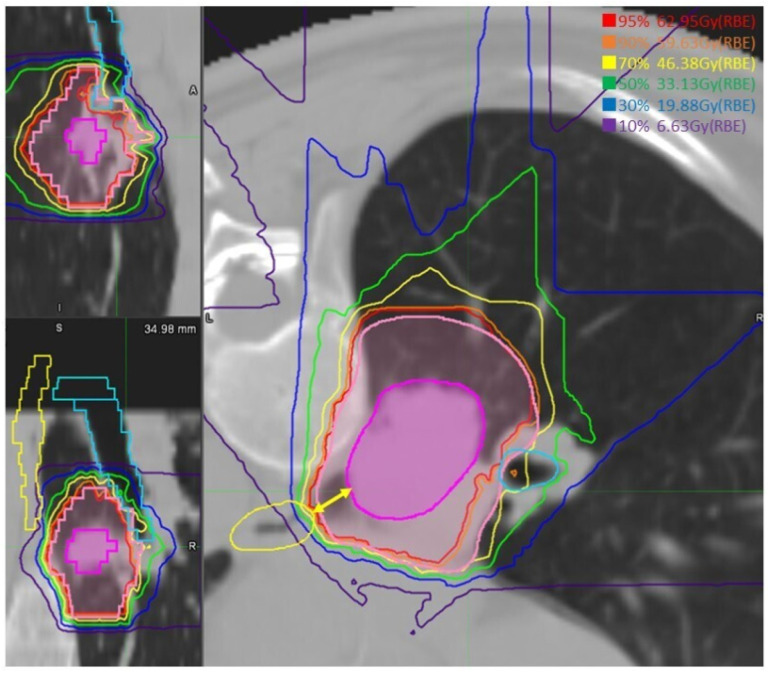
Dose distribution for one case with the highest maximum dose to the esophagus. The point dose was high at 60.7 Gy, while the D1cc dose was 5.4 Gy. The minimum distance between the GTV (magenta) and the esophagus, shown by a yellow arrow, was 8.6 mm.

**Figure 3 cancers-16-00933-f003:**
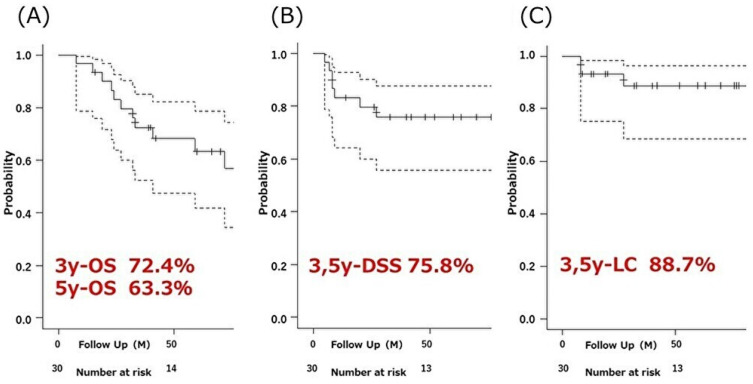
Kaplan–Meier curves of (**A**) overall survival (OS), (**B**) disease-specific survival (DSS), and (**C**) local control (LC) in the whole cohort.

**Table 1 cancers-16-00933-t001:** Patient characteristics (n = 30).

Age (y)	median (range)	75 (55–85)
Gender	Male/Female	21/9
ECOG performance status	0/1	24/6
Operability	Yes/No	10/20
Severity of COPD: GOLD score		1/2/3/4
Hypertension	Yes/No	13/17
Diabetes mellitus	Yes/No	5/25
Cardiovascular disease	Yes/No	6/24
Histology	Adeno	15
	Squamous	13
	Others	2
c-Stage (UICC 8th)	1A	8
	1B	12
	2A	6
	2B	4
Most constrained OAR	Main bronchus	4
	Lobar bronchus	23
	Others	3
Baseline pulmonary function	FEV1 (liter, median)	1.7 (0.4–2.8)
	FEV1% (median)	73.8 (28.3–90.0)
	VC (liter, median)	2.6 (1.2–4.5)
	VC% (median)	87.8 (65.6–138.1)
Age (y)	median (range)	75 (55–85)

Abbreviations. ECOG, Eastern Cooperative Oncology Group; COPD, chronic obstructive pulmonary disease; GOLD, Global Initiative for Chronic Obstructive Lung Disease; c-Stage, clinical stage; OAR, organ at risk; FEV1, forced expiratory volume in 1 s; VC, vital capacity.

**Table 2 cancers-16-00933-t002:** Irradiation doses for major OARs.

OARs	Dose Parameters	Total Dose (Median [Range])
Lung	V5_%	15.2 (10.2–24.1)
	V20_%	10.4 (4.1–18.2)
	MLD_Gy	5.6 (2.7–10.9)
PBT	Dmax_Gy	65.6 (3.8–71.2)
	D0.2cc_Gy	58.8 (2.8–70.7)
	D0.5cc_Gy	52.8 (2.7–69.2)
	D1cc_Gy	39.5 (2.4–68.6)
	D2cc_Gy	10.0 (0.2–51.6)
Esophagus	Dmax_Gy	9.9 (1.8–60.7)
	D0.2cc_Gy	4.2 (1.4–37.5)
	D0.5cc_Gy	3.6 (0.8–30.9)
	D1cc_Gy	3.0 (0.7–23.5)
Spinal cord	Dmax_Gy	2.4 (0–18.2)
	D0.2cc_Gy	1.5 (0.0–17.4)

Abbreviations. PBT, proximal bronchial tree; V5 (20), percentage of the volume of an organ receiving 5 (20) Gy; MLD, mean lung dose; Dmax, maximum point dose; D0.2cc/0.5cc/1cc/2cc, dose to the hottest 0.2cc/0.5cc/1cc/2cc of OAR.

**Table 3 cancers-16-00933-t003:** Factors Associated with OS.

		No. of Patients	OS	
			5y (%)	*p*-Value
Age	≤75	14	85.1	0.047 *
	>75	16	46.4	
Gender	Male	21	50.3	0.18
	Female	9	88.9	
PS	0	24	73.8	0.088
	1	6	33.3	
Severity of COPD	GOLD1,2	24	67.7	0.16
	GOLD3,4	6	41.7	
c-Stage	1	20	74.7	0.75
	2	10	40.5	
Histology	Squamous	13	36.9	0.015 *
	the others	17	84.7	

Abbreviations. ECOG, Eastern Cooperative Oncology Group; c-Stage, clinical stage; OS, overall survival; *, *p* < 0.05.

**Table 4 cancers-16-00933-t004:** Acute and late adverse events of all patients (n = 30).

Acute Adverse Events	Grade2	Grade3	Grade4–5
Pneumonitis	3 (10.0%)	0	0
Dermatitis	0	0	0
Esophagitis	0	0	0
**Late Adverse Events**	**Grade2**	**Grade3**	**Grade4–5**
Pneumonitis	2 (6.7%)	2 (6.7%)	0
Dermatitis	0	0	0
Esophagitis	0	0	0
Chest wall pain	0	0	0
Bronchial stenosis	1 (3.3%)	0	0

**Table 5 cancers-16-00933-t005:** Previous reports of ablative radiation therapy for central lung tumors.

Author (Year)	Disease	No. of pts	Prescription (Gy/Fr)	BED10 (Gy)	Follow (m)	LC (%)	OS (%)	Toxicity		Details of G5 Toxicity
								G ≥ 3 (%)	G5 (%)	
Timmerman (2006) [13]	T1–2N0	70	60–66/3	180–211.2	18	95 (2 y)	54.7 (2 y)	20	8.6	1 hemorrhage, 1 peripheral effusion 4 pneumonias
Milano (2009) [39]	T1–3N0 or M1	53	30–63 (2.5–5 Gy/Fr)	39–82.5	10	73 (2 y)	44 (2 y)	11 (n = 45)	8.9	3 pulmonary declines, 1 hemoptysis
Song (2009) [40]	T1–2aN0	32	40–60/4–6	80–180	27	85.3 (2 y)	38.5 (2 y)	33.3 (n = 9)	11.1	1 hemorrhage
Haasbeek (2011) [41]	T1–3N0	63	60/8	105	35	92.6 (3 y)	64.3 (3 y)	20.6	14.3	9 cardiopulmonary causes
Rowe (2012) [42]	T1–2N0 or M1	47	50/4	76 (60–151.2)	11	94 (2 y)	N/R	10.6	2.2	1 hemorrhage
Modh (2014) [15]	T1–4N0 or M1	125	30–60/2–5	85.5 (43–180)	17	79 (2 y, BED ≥ 80)	64 (2 y, BED ≥ 80)	8	6	1 hemorrhage
Tekatli (2015) [43]	stage 1A-4	80	60/8	105	45	62 (2 y), 53 (3 y)	N/R	13.8 (n = 78)	7.5	3 respiratory failures, 2 hemorrhages, 1 sudden death
Aoki (2018) [14]	T1–3N0 or M1	35	56/7	100.8	13.1	96 (2 y)	40.4 (2 y)	26	5.7	1 hemorrhage, 1 pneumonia
Bezjak (2019) [15]	T1–2N0	71	57.5–60/5	123.6, 132	33	89.4, 87.9 (2 y)	67.9, 72.7 (2 y)	21	5.6	3 hemorrhages, 1 esophageal ulcer
Lindberg. K (2021) [29]	≤5 cm (NSCLC/LM)	65	56/8	95.2	24	83 (2 y)	58 (2 y)	39.3	17.9	8 hemorrhage, 1 pneumonitis, 1 fistula
Current study	T1–3N0	30	68.4/12	107.4	42	88.7 (3 y, 5 y)	72.4 (3 y), 63.3 (5 y)	6.7	0	

Abbreviations: pt, patients; BED, biological equivalent dose; LC, local control; OS, overall survival; G, grade; Fr, fraction; y, years; n, number; N/R, data not reported; NSCLC, non-small cell lung cancer; LM, lung metastases.

## Data Availability

The data presented in this study are available on request from the corresponding author.

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
