# Peer review of "Long-Term Outcomes of Ablative Carbon-Ion Radiotherapy for Central Non-Small Cell Lung Cancer: A Single-Center, Retrospective Study"

_cancers, 2024, doi:10.3390/cancers16050933_

Round 1
Reviewer 1 Report
Comments and Suggestions for Authors
This is a small retrospective series of 30 patients, included in a time period of 13 years and treated with carbon ions in Chiba, Japan. All patients had so-called central T1-T2N0M0 lung cancer.
The results appear to be superior in terms of severe toxicity than what has been published with photon SBRT, although the selection bias and the fact that this is a retrospective series thus possibly underestimating toxicity is a concern. Indeed, the allowed dose to the main bronchi is expressed as a small Dmax volume. As in nearly all photon series, also here, no details are given about the volume and doses to each OAR. Simply reporting a Dmax is not sufficient. Only then we may have an idea what exactly what has received which dose on which volume. This should be given by the authors.
It is nevertheless appealing to speculate that the physical characteristics of carbon ions relate to the sharp dose gradients and hence to the low toxicity rate.
The table does not include important Scandinavian prospective and even randomized trials studies such as HILUS etc., in which SBRT is given to central tumors, ultracentral tumors and where protracted radiotherapy is compared to SBRT for similar central tumors as in this series. The conclusion of the randomized study is that protracted schedules do offer a good alternative to SBRT when the latter is considered to be too risky. This is important information for the clinician and the patient that should be added.
Author Response
Thank you very much for taking the time to review this manuscript.
Please see the attachment.

Reviewer 2 Report
Comments and Suggestions for Authors
The manuscript focuses on the high-dose carbon-ion radiotherapy (CIRT) may be a safe and effective intervention for central non-small cell lung cancer (NSCLC), as dose reduction to intrathoracic organs at risk (OARs) reduces the risk of fatal adverse events (AEs), which helps patients' decision-making the best treatment for central NSCLC. To render the manuscript suitable for publication to Cancers, some parts of the manuscript need to be modified.
1. Tables 1-4, the quality was not good enough (difficult to understand) and should be improved.
2. Please provide more information/and or discussion about the CIRT, proton and photon SBRT for central NSCLC, particularly the side effect to OARs. Please provide example to support your discussion.
3. In abstract, we reviewed 30 patients who 36 had received CIRT at 68.4 Gy in 12 fractions for central NSCLC in 2006–2018. But, in materials and method, Patients with central lung tumors who had received CIRT with 68.4 Gy in 12 Fr at 75 QST Hospital in Chiba, Japan, between July 2006 and September 2019 were retrospectively 76 analyzed. Please make adjustments 2018 or 2019.
4. Please provide the value of RBE and LET for carbon ions in the manuscript.
Comments on the Quality of English LanguageMinor editing of English language required.
Author Response

(The authors gave the same response as above.)

Reviewer 3 Report
Comments and Suggestions for Authors
Thank you for submitting this interesting and informative manuscript to Cancers. I was pleased to receive it as a reviewer.
While your manuscript provides valuable insights into an important clinical topic, there are certain areas that could be refined to further enhance the quality and impact of the work. Here are some respectful suggestions that could potentially improve the paper if you choose to implement them:
Introduction
- Consider expanding on the underlying mechanisms of CIRT that motivate its potential advantages over photon therapy (biological effects, physics) to strengthen the rationale for the present study.
- Consider summarising prior early phase dose escalation data that formed the basis for the current regimen to showcase the deliberate buildup supporting this study.
Methods
- Consider providing more details on patient baseline characteristics (comorbidities, pulmonary function tests) for readers to assess generalizability.
- Consider providing examples of CIRT treatment planning objectives and constraints for OARs to showcase the safety-centric design.
Results
- Consider adding a table summarizing key Grade 2+ adverse events for comprehensive toxicity reporting.
- Consider presenting outcomes stratified by tumour stage to look for signals of heterogeneity in subgroup response.
Discussion
- Consider commenting on the potential influence of immunotherapy sequencing on outcomes given the enrolment period.
- Consider further discussing whether the findings support application in the curative or salvage setting for central lung tumours.
Conclusions
- Consider providing specific implications of the findings in terms of guiding patient selection, prescription guidelines, and future research for enhanced impact on the field
Overall, these suggested edits could further maximize this paper’s influence on shaping ongoing dialogue regarding the evolving role of CIRT versus photons for delivering ablative doses to central tumours in the lung.
Author Response

(The authors gave the same response as above.)

Round 2
Reviewer 1 Report
Comments and Suggestions for Authors
The main text has changed appropriately, whereas the abstract has not. It should clearly be stated what the dosimetry is, albeit in short. This is crucial as most people only read the abstract.
Author Response
Thank you very much for your prompt review of the revised manuscript and again for your suggestions. Please see the attachment.
